# Potential Associations among Bioactive Molecules, Antioxidant Activity and Resveratrol Production in *Vitis vinifera* Fruits of North America

**DOI:** 10.3390/molecules27020336

**Published:** 2022-01-06

**Authors:** Champa Wijekoon, Thomas Netticadan, Yaw L. Siow, Ali Sabra, Liping Yu, Pema Raj, Suvira Prashar

**Affiliations:** 1Agriculture and Agri-Food Canada, Winnipeg, MB R6M 1Y5, Canada; thomas.netticadan@agr.gc.ca (T.N.); chris.siow@agr.gc.ca (Y.L.S.); ali.sabra@agr.gc.ca (A.S.); liping.yu@agr.gc.ca (L.Y.); pemaraj@gmail.com (P.R.); suvira.prashar@agr.gc.ca (S.P.); 2Canadian Centre for Agri-Food Research in Health and Medicine, Winnipeg, MB R3C 1B2, Canada; 3Department of Physiology and Pathophysiology, University of Manitoba, Winnipeg, MB R3T 2N2, Canada

**Keywords:** bioactive molecules, antioxidant activity, grapes, resveratrol, stilbene synthase

## Abstract

Grapes (*Vitis vinifera* L.) are rich in bioactive molecules contributing to health benefits. Consumption of grapes is linked to reduced incidence of cardiovascular diseases. Studies on table grape cultivars are limited although much attention in research was focused on the wine industry. Bioactive effects of grapes as anti-inflammatory, anticarcinogenic, cardioprotective, vasorelaxant, phytoestrogenic and neuroprotective have also been reported. For example, resveratrol is a natural food ingredient present in grapes, with high antioxidant potential. Here we conducted an exploratory study to investigate bioactive molecules, antioxidant activity and the association between constitutive stilbene synthase (STS) gene expression and the resveratrol biosynthesis in selected table grape varieties in North America. The phenolic compounds, fatty acid composition and antioxidant activity of four grape varieties were compared. Red Globe variety was rich in unsaturated fatty acids as well as phenolic compounds such as caffeic acid, quercetin and resveratrol. Meanwhile, the constitutive expression of grape stilbene synthase gene was higher in Flame and Autumn Royal where resveratrol content of these cultivars was relatively low compared to the Red Globe variety. This study shows the potential links in grape antioxidant activity and resveratrol production, but more studies are necessary to show the association.

## 1. Introduction

*Vitis vinifera* fruits (grapes) are a rich source of polyphenolic compounds that attribute to the health benefits of their consumption [1]. Fresh (table) grapes account for less than 12% of the world’s total grape production. China is the world’s largest producer and consumer of fresh table grapes in the marketing year of 2020/2021 and the United States, forecasted to produce 871000 tons in 2021, gained the 5th place in table grape production according to the 2020/2021 global table grape report [2]. It is believed that first European colonists introduced European grape to the North America but their first attempts to grow it resulted in failure due to its sensitivity to cold temperatures [3]. The initial stock of the North American grapes, later called variety ‘Catawba’ was discovered in a forest near the Catawba River in North Carolina in 1801 [4]. It is also considered that Spanish explorers brought the European cultivars and established vineyards in California in the late 18th century introducing them in North America [5]. Due to selection, accidental crosses or breeding processes between the domesticated grapes and native wild grapes, some American cultivars were developed. For example, native grape cultivars such as *V. labrusca* and *V. aestivalis* were used to develop Concord and Norton grapes, respectively. Seedless grape varieties were also developed to fulfil table grapes demand based on consumer preference, but researchers are now discovering that many health benefits of grapes may actually come from the seeds as well due to their enriched phytochemical composition and bioactive molecules [6,7,8].

Bioactive polyphenolic compounds in grapes include flavonoids and stilbenes [9]. In addition to polyphenolic compounds, grape seed oil contains also healthy fatty acids particularly the unsaturated fatty acids, such as linoleic acid and oleic acid that increase the nutritive value of grape oil when used in food or dietary supplements [1]. Studies have shown that grape seed oil exhibits anti-inflammatory, antioxidant, cardio-protective and anticancer properties, which could be due to the occurrence of linoleic acid, tocopherol, carotenoids, phytosterols in addition to some polyphenolic compounds such as proanthocyanidins, resveratrol and quercetin [10]. 

To date, much attention has been given to wine grapes; however, studies focusing on table grape bioactive composition and their underlying molecular mechanisms are very limited. Stilbene resveratrol, a naturally occurring phytochemical in grapes, is a potent antioxidant and may play a role in protecting against cardiovascular diseases [1]. In addition, studies have shown that whole grape berry extract might be more efficient in exerting the health benefits such as showing cytotoxic activity against lung carcinoma cells, breast cancer and human liver cancer cells than a pure phytochemical compound [11]. 

Stilbenes, including resveratrol, are synthesized via the phenylpropanoid pathway [12]. Phenylalanine ammonia-lyase (PAL) the first enzyme in this pathway, and Stilbene synthase (STS) or resveratrol synthase condenses malonyl-CoA and cumaryl-CoA to form resveratrol [13]. It has been suggested that the selective enhancement of expression of individual *PAL* and *STS* genes involved in resveratrol biosynthesis [14]. Resveratrol is present in many plants including medicinal plants and in edible plant sources including soft fruits such as grapes, lingonberries, and blueberries, as well as in nuts such as peanuts and pistachios [15,16]. Resveratrol is a small molecule with a molecular weight of 228.24 g/mol, and it contains two phenol rings that are connected by an ethylene bridge. It occurs as two distinct isomers, trans-resveratrol and cis-resveratrol [16]. The isomerization from trans resveratrol to cis-resveratrol occurs when trans-resveratrol is exposed to a high pH (more than 11), it can also occur when exposed to sunlight and UV radiation of 254 nm or 366 nm [16,17]. Both isomers possess lipophilic properties; however, trans-resveratrol is the more prevalent form. Research conducted thus far indicates that trans-resveratrol is the more biologically active form as it has been studied more than cis-resveratrol because of its higher stability. Nevertheless, trans-resveratrol and cis-resveratrol have been reported to have different effects.

In this study, we investigate the bioactive composition and the antioxidant activity of the selected fresh berries of North American table grape cultivars (Figure 1) present in market with the extended study of association of constitutive *STS* gene expression on resveratrol production. The purpose of this study is to understand the changes in bioactive composition in fresh *Vitis vinifera* table grape varieties in Market and to identify potential molecular mechanisms underlying the biosynthesis of the potent bioactive compound resveratrol.

## 2. Results

### 2.1. Assessment of Phenolic Bioactive Compounds in Table Grapes

High performance liquid chromatography enabled the identification of diverse chemical groups including cinnamic acid derivatives, hydroxybenzoic acid, flavan-3-ols, flavonols, flavones, stilbenes, and anthocyanins. Tested grapes showed a variation in bioactive compounds based on the variety. For instance, the black colored variety (Autumn Royal) had comparatively higher caftaric acid content, while the Sweet Scarlet had significantly lower caftaric acid content compared to other varieties. In addition, caffeic acid content in the variety Red Globe, was nearly four times higher compared to Autumn Royal (Table 1).

In general, Autumn Royal had the highest and twice the amounts of cinnamic acid derivatives compared to Sweet Scarlet variety. The opposite trend was observed for the flavan-3-ols, where the total content of Sweet Scarlet was more than twice the amount of Autumn Royal. Sweet Scarlet variety has high contents of chlorogenic acid, caffeic acid and anthocyanins content compared with other seedless varieties while it has the highest levels of protocatechuic acid, ferulic acid, catechin, epicatechin and isoquercetin levels among all the studied varieties. However, Flame is comparatively rich in cinnamic acid derivatives to the Sweet Scarlet variety. In addition, Autumn Royal showed significantly high epigallocatechin and epicatechin gallate content (Table 1).

The flesh of the seeded variety (Red Globe) showed significantly higher quercetin, myricetin contents compared to other varieties. In addition, iso-quercetin was higher in both Sweet Scarlet and Autumn Royal. Interestingly, Red Globe showed the highest resveratrol content and it was more than two times higher compared with seedless varieties (Table 1 and Figure 2A,B).

### 2.2. Comparison of Fatty Acid Composition with Seedless Varieties and GRAPE Seeds/Seed Oil

Percentage of fatty acid methyl esters (FAMEs) showed a variation among the grape varieties tested (Table 2). For example, Methyl linoleate (C18:2) was the predominant constituent in the seeds of Red Globe variety as well the flesh of the three seedless varieties. In this regard, the content of C18:2 in seed oils was at least more than five times higher than the C18:2 content in the seedless varieties. Other major compounds, such as Methyl linoleate (C18:3) was found in all varieties with high abundance. Compared to the total profile of unsaturated (USFA) and saturated fatty acids (SFA) in seedless varieties, the ratio of USFA/SFA was two times higher in the Red Globe seeds (Table 2 and Figure 2C).

### 2.3. Analysis of Antioxidant Activity Assay in Seeds in and Seedless Grapes

The 2,2′-azino-bis3-ethylbenzothiazoline-6-sulfonic acid (ABTS) antioxidant activities of the grape varieties are given in Figure 1. Flame (seedless), Autumn Royal (seedless), Sweet Scarlet (seedless), Red Globe (pitted) and Red Globe seeds showed high antioxidant activity (measured as to equivalent Trolox units). There was no significant variation in the level of antioxidant activity between different varieties of grape extract in this study (Figure 3).

### 2.4. Expression of Stilbene Synthase Gene and Resveratrol Production

Relative expression of STS gene tested with qRT-PCR analysis showed that the STS gene was constitutively expressed in all the grape varieties tested. Amplified product distinct for STS transcript was 137 bp. Constitutive Tubulin gene transcript was 86 bp. Interestingly, Flame and Autumn Royal varieties showed the higher relative gene expression values compared with other two varieties (Figure 4).

## 3. Discussion

Previous evidence supports that food rich in a wide variety of bioactive compounds including phenolics and polyunsaturated fatty acids have demonstrated blood pressure lowering potential [18]. It has also been suggested that fruit-derived phenolic compounds favorably affect four risk factors of cardiovascular diseases: platelet aggregation, elevated blood pressure, vascular dysfunction and hyperlipidemia [19]. The grapes examined in this study were rich in phenolic acids which have broader role as anti-inflammatory, antidiabetic, anti-cancer, antiapoptotic, antiaging, hepatoprotective, neuroprotective, radioprotective, pulmonary protective, hypotensive effect, and antiatherogenic effects [20]. High contents of bioactive molecules and the antioxidant activity of Red Globe, Sweet Scarlet, Autumn Royal may contribute to the health benefits as suggested by researchers [1].

Seeds containing grape varieties may have more cardio-protective benefits due to additional contents of phenolic compounds such as proanthocyanins and unsaturated fatty acids [20]. For example, Red Globe seeds can be considered a rich source of oil with nearly 70% of linoleic acid (Table 2), a fatty acid with high biological activity [1], which could have some implications in nutrition, cosmetics or drug industries. This variety is also rich in quercetin and resveratrol (Table 1) which are potent antioxidants and have been suggested to have a role in protecting against cardiovascular diseases [21,22].

Fruit of *V. vinifera* is a valuable fruit with the good antioxidant capacity [23]. In this study, we observed that all varieties (Flame, Autumn Royal, Sweet Scarlet, Red Globe) are potent antioxidants with significant capacity to scavenge ABTS radical cation (Figure 2). Previously, grape juice has been shown to have antioxidant activity and inhibit oxidation of LDL [24]. Grape juice was able to prevent high-fat diet induced rise in thiobarbituric acid-reactive substances and decrease in superoxide dismutase activity [25]. Short-term ingestion of purple grape juice improved flow mediated vasodilation assessed by high-resolution brachial artery ultrasonography and reduced LDL oxidation in coronary artery disease patients [26]. In addition, healthy adults administered with 10 mL CGJ · kg^−1^·d^−1^ also showed high antioxidant capacity and less LDL oxidation similar to 400 IU alpha-tocopherol, which is a potent antioxidant [27]. In vivo studies are warranted to examine the antioxidant capacity of Flame, Autumn Royal, Sweet Scarlet, Red Globe at different doses and treatment durations. Such studies may provide valuable information about the therapeutic potential of various grape varieties in the setting of oxidant imbalances and associated diseases.

Despite the health benefits of particular varieties of grape may stand more, consumer preferences, taste, flavor, aroma, and other organoleptic characteristics may influence the choice of a particular variety for consumption. Based on the results of the study, tested table grapes in Winnipeg market are rich source of bioactive molecules with a link to the high antioxidant activity (Table 1 and Table 2 and Figure 3).

Based on the results of the in vitro health parameters assessed (antioxidant activity and metabolite profile) with the different grape varieties in current study, it may be important to explore the potential of the studied grapes varieties in preventing the development of disease, in vivo. In this context, members in our group have previously reported that administration of California grape powder lowered severe hypertension and prevented cardiac hypertrophy (abnormal enlargement of the heart) in the spontaneously hypertensive rat (SHR), an animal model of hypertension [28] hypertension is one of the major risk factors of diseases including cardiovascular diseases. Another study by our members also revealed that recovery of heart function with administration of resveratrol in the SHR [29], and in the high fat fed rat [30] in an animal model of obesity. Obesity is another major risk factor for diseases including cardiovascular disease. Thus, future studies could be designed to examine the effects of the grape varieties (in the current study) utilizing established animal models of risk factors of diseases. As referred to above, the SHR is considered to be the most commonly used animal model of hypertension, while the high-fat fed rat is one of the most commonly used animal models of obesity. The high fat, high sugar fed rat is also another commonly used animal model of diabetes which is another major risk factor of the diseases. Red Globe has the highest content of stilbenes compared with other three varieties; however, relatively lower STS gene transcript levels were observed compared to Flame and Autumn Royal varieties (Table 1, Figure 3). Therefore, relative constitutive *STS* gene expression levels may not have a major role in resveratrol biosynthesis. It has been demonstrated that ‘Red Globe’ is a light-depended grape variety where light-induced anthocyanin synthesis process may have been involved with multiple transcription factors regulating different functional genes [31]. Similarly in the process of resveratrol synthesis, multiple genes may be involved, and the constitutive expression of *STS* gene alone may not directly associate with the impact.

## 4. Materials and Methods

### 4.1. Plant Materials

Fruits of grape varieties such as Flame Seedless, Autumn Royal, Sweet Scarlet and Red Globe (Figure 1) were purchased from retail market in Winnipeg, MB, Canada. All grapes were stored at −80 °C. Experiments were repeated three times with three replications each with new group of grapes. Samples used included randomly picked and pooled normal healthy grapes from each grape lot.

### 4.2. Antioxidant Activity Assay Analysis

2,2′-azino-bis3-ethylbenzothiazoline-6-sulfonic acid (ABTS) antioxidant assay was used to determine the total antioxidant capacity of grape varieties such as Flame, Autumn Royal, Sweet Scarlet and Red Globe following manufacturer’s instructions (AMSBIO, Cambridge, MA, USA). The assay measured ABTS radical cation formation induced by metmyoglobin and hydrogen peroxide. Trolox [6-Hydroxy-2,5,7,8-tetramethylchroman-2-carboxylic acid], a water-soluble vitamin E analog, served as a positive control inhibiting the formation of the radical cation in a dose dependent manner. The antioxidant activity was normalized to equivalent Trolox units to quantify the composite antioxidant activity present [32].

### 4.3. Chemicals and Standards

Standards, such as caftaric acid, chlorogenic acid, *p*-coumaric acid, ferulic acid, rutin, quercetin, iso-quercetin, myricetin, kaempferol, genistein, resveratrol, catechin, epicatechin, epigallocatechin, epicatechin gallate, heptadecanoic acid, as well as chemicals, such as n-hexane and hydrochloric acid were purchased from Millipore-Sigma (St. Louis, MI, USA). Methanol, Acetonitrile, orthophosphoric acid were HPLC grade and were obtained from Fisher Scientific. Water used in analysis was high purity water obtained from a Millipore purification system.

### 4.4. Extraction and Analysis of Phenolic Compounds by HPLC

Extraction of phenolic compounds from grape berries was adopted from [33] with modifications. Whole grapes were freeze-dried and stored at −80 °C. A composite sample was ground in mortar using pestle and one gram from each variety was weighed into 50 mL centrifuge tubes. Ten mL from 80% aqueous ethanol was added and sonicated for 1 h at 40 °C. Extracts were filtered using Whatman #4 filter paper into new 15 mL centrifuge tubes, then concentrated under vacuum using Vacufuge Plus (Eppendorf) at 60 °C for 2 h. Tubes were centrifuged at 14,000 rpm for 20 min, and the supernatant was transferred to new tubes and placed on dry bath incubator for 4 h at 60 °C until volume reduced. HPLC-grade methanol was added, and volume was adjusted to 3.0 mL. Extracts were filtered using 0.2 µm Nylon syringe filters (Fisher Scientific, Lenexa, KS, USA) into HPLC vials prior to analysis. Analysis was performed using Dionex 3000 Ultimate (Thermo Scientific, Lenexa, KS, USA) on C18 reverse phase column (Acclaim 120, 4.6 × 250 mm, 5 µm). Separation of compounds was based on a gradient elution utilizing acidified water (0.1% phosphoric acid) as mobile phase A and acetonitrile as mobile phase B. Gradient elution started with 10% B increasing to 90% in 35 min and then equilibrate to initial concentration before next injection. Injection volume was 20 µL. Compounds were acquired at 306 for stilbenes, 325 nm for hydroxycinnamic acids, 205 nm for flavonols and flavan-3-ols, and 520 nm for anthocynanins. Identification of compounds was completed by comparing the retention times (Rts) of individual compounds to the Rts of corresponding standards and by spike addition method. Quantification of compounds was completed by making serial dilutions from each standard and establishing calibration curves covering at least 5 concentration points. The regression equation was used to quantify each compound based on the detector response (peak area) and the results expressed as µg/mg dry weight.

### 4.5. Extraction and Analysis of FAMEs by GC-MS

Fatty acid analysis was performed using a one-step extraction-methylation protocol adapted from [34] with some modifications. About 200 mg of powdered plant material was weighed and placed in 12 mL Pyrex tubes and 40 µL (1.25 mg/mL) of the internal standard (C17:0) was added. Two mL of 3 N methanolic-HCl was added to each sample, capped and vortexed for 30 s. Tubes were incubated in a dry bath at 80 °C for 45 min. After cooling down, fatty acids methyl esters (FAMEs) were extracted by the addition of 4 mL hexane and vortexed for 30 s, then left overnight under dark conditions. The hexane layer (top layer) was transferred to new glass disposable tube and evaporated to dryness using Vacufuge Plus sample concentrator (Eppendorf AG, Hamburg, Germany). The extract was resuspended in 1 mL hexane and vortexed for 30 sec and transferred to GC vials prior to analysis. Analysis of FAMEs was completed using Bruker 436-GC equipped with EVOQ-TQ-MS (Bruker Daltonics, Germany). Compounds were separated on Rt-2560 capillary column (100 m × 250 μm × 0.20) (Restek, Bellefonte, PA, USA). Helium was used as the carrier gas with a flow rate of 0.8 mL/min. The oven was programmed with initial temperature 100 °C for 4 min ramping up to 250 °C at a rate of 3 °C/min and holding for 8 min. Ion source temperature was 230 °C, while the temperature for the transfer line was 180 °C. The mass spectra was acquired from 50 to 500 amu in a full scan mode using MSWS software. Identification of FAMEs was based on comparing the retention times with the retention times of standard mix (Supelco 37 Component FAME Mix, Millipore Sigma, Saint Louis, MO, USA) and searching NIST library. Quantification of compounds was based on using an internal standard method.

### 4.6. RNA Extraction and cDNA Synthesis

Frozen samples were ground and powdered using liquid Nitrogen. Total RNA was extracted from the grapes representing three biological samples from each variety using the Sigma Spectrum Plant Total RNA kit (Sigma-Aldrich Canada Co. 2149 Winston Park Drive, Oakville, ON, Canada) and reverse transcribed using the Superscript VILO cDNA synthesis kit (Thermo Fisher Scientific, Lenexa, KS, USA). Prior to cDNA quality and quantity of RNA was checked using NanoDrop (Thermo Scientific NanoDrop Products). Complementary DNA was assessed for the presence of stilbene synthase sequence (AB046375.1) and the constitutive tubulin transcript (AT1G75780.1) using Gene Amp Fast PCR Master mix and primers SBSF 5′ GAGTGTGTGCTATTAGGTGAGGC and SBSR 5′GTCAAATGAAAGGAATATTGGTTACTG yielding 137bp fragment. Beta-tubulin primer set TuF 5′ TGAACCACTTGATCTCTGCGACTA and TuR 5′CAGCTTGCGGAGGTCTGAGT produced 86 bp amplicon (Figure 4). A thermocycling regime of 95oC for 1m followed by 45cycles of 95 °C for 05 s, 60 °C for 25 s, 72 °C for 15 s and 72 °C for 1 min was used. PCR products were validated by running on an agarose gel.

### 4.7. Quantitative Real Time (qRT) PCR Analysis

Quantitative real-time PCR was performed using the StepOne Plus Realtime Polymerase Chain Reaction System (Applied Biosystems, Foster City, CA, USA).

Reactions were performed in a final volume of 10 μL using 1x iTaq Universal SYBR Green Super Mix. (Bio-Rad, Hercules, CA, USA), 100 ng cDNA, and 750 nM primer designed to anneal to a region of the coding sequence that was distinct for stilbene synthase 137 bp product. Primers TuF and TuR (described above) were used to amplify a 86 bp fragment of the constitutively expressed beta-tubulin transcript, which was utilized as an internal control. Thermal parameters for amplification were as follows: 10 min at 50 °C and 1 min at 95 °C, followed by 40 cycles of 95 °C for 30 s and 60 °C for 15 s. Primer-pair specificity was validated for each RT-qPCR experiment using a dissociation curve, which demonstrated a single amplicon for each of the targeted transcripts. All samples were tested in triplicate. The fold change expression of the target gene stilbene synthase was measured relative to the internal control gene (βTubulin) using comparative Ct method.

### 4.8. Statistical Analysis

Quantitative data were subjected to one-way ANOVA using JMP 16 statistical program. (SAS, Cary, NC, USA). Tukey’s HSD test was used to compare the significant differences between varieties at ≤0.05 level. Values accompanied by different letters were significantly different.

## 5. Conclusions

Though table grape consumption is known to have health benefits, each variety has its own unique bioactive compound profile. Therefore, it is challenging to attribute health benefits to a particular compound or a group of compounds, as well as to recommend a particular cultivar that provides all spectrum of benefits to all consumers. It seems that Flame Seedless variety may have less nutritional value, as its composition of bioactive markers was significantly lower than other varieties. On the other hand, the seeded variety (Red Globe) could be considered a significant source of anthocyanins, flavones, flavonols, and stilbenes (resveratrol). This may highlight the importance of consuming grape varieties with seeds as a functional food. The potential future animal studies suggested in discussion could assess in vivo organ structure and function to complement the examination of biochemical and molecular markers of disease in the blood and tissue (of the animals studied).

## Figures and Tables

**Figure 1 molecules-27-00336-f001:**
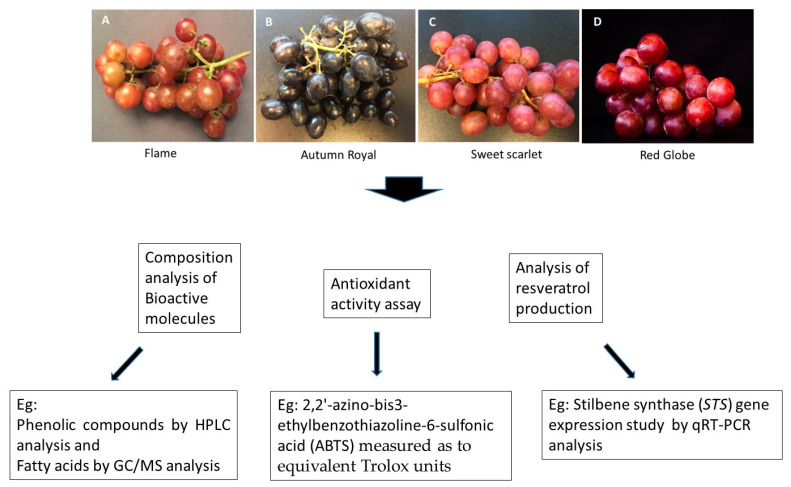
Schematic representation of the methods used in this study. Commercial table grape varieties tested from the Winnipeg market are (**A**) Flame, (**B**) Autumn Royal, (**C**) Sweet Scarlet and (**D**) Red Globe. Bioactive composition was analyzed using HPLC and GC/MS. Antioxidant activity was determined for all 4 varieties. Constitutive stilbene synthase gene expression on resveratrol production was also studied using quantitative real-time PCR (qRT-PCR) method.

**Figure 2 molecules-27-00336-f002:**
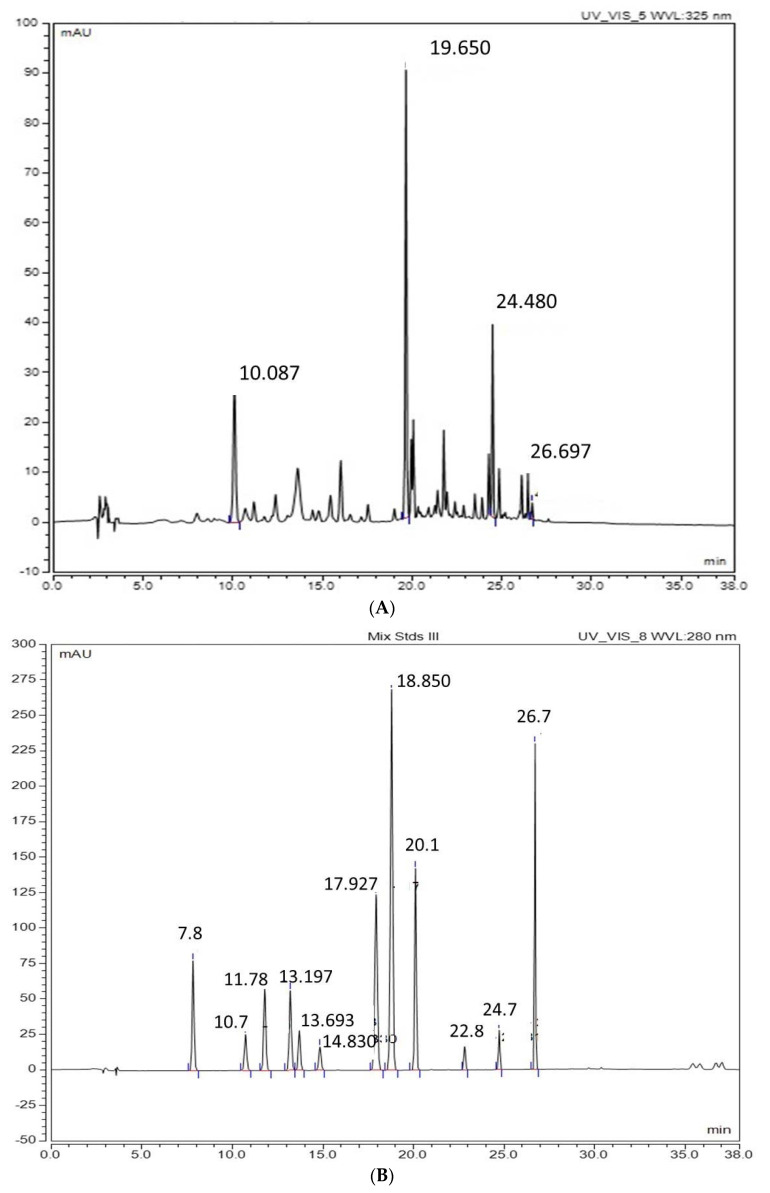
Chromatograms obtained from HPLC and GC/MS. (**A**) Chromatogram of phenolic compounds identified in Red globe grape extract at 280 nm. (**B**) HPLC chromatogram of a mix of phenolic compound standards detected at 280 nm. Rt 7.8 Protocatechuic acid; Rt10.7 Catechin; Rt 11.78 4-Hydroxybenzoic acid; Rt 13.197 Vanillic acid; Rt 13.693 Epicatechin; Rt 14.830 3-Hydroxybenzoic acid; Rt 17.927 Vanillin; Rt 18.850 *p*-Coumaric acid + syringaldehyde; Rt 20.10 Ferulic acid; Rt 22.8 Benzoic acid; Rt 24.7 Salicylic acid; Rt 26.7 Genistein. (**C**) GC/MS total ion chromatogram of the fatty acid methyl esters extracted from the grape varieties (Flame, Autumn Royal, Sweet Scarlet, Red Globe pitted, and seeds).

**Figure 3 molecules-27-00336-f003:**
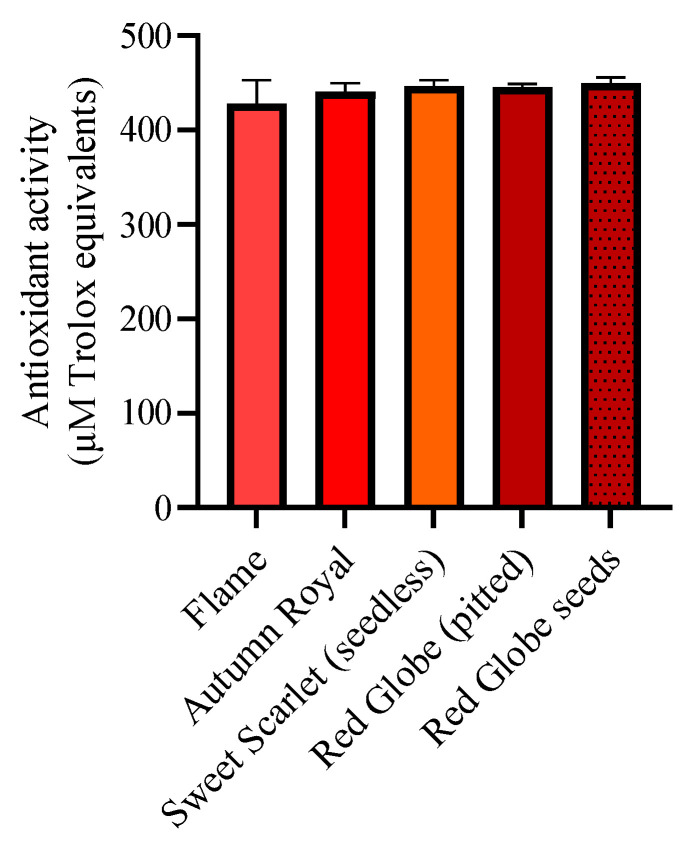
The antioxidant activity of the four grape varieties, Flame, Autumn Royal, Sweet Scarlet (seedless), Red Globe (pitted) and seeds. All the tested grape varieties including their seeds showed high antioxidant activity assay without significant changes from each other. The results are expressed as SE ± (n = 3).

**Figure 4 molecules-27-00336-f004:**
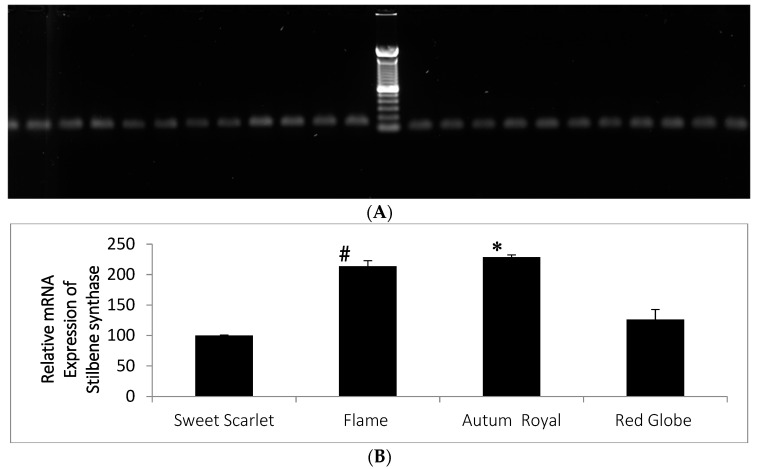
Stilbene synthase (*STS*) gene expression analysis in tested four varieties of table grapes. (**A**). Amplified products of *STS* (137 bp) and *beta-tubulin* (86 bp) by PCR analysis. (**B**). Relative expression of *STS* gene in different varieties of table grapes found in Winnipeg market. (The results are expressed as SE ± (n = 3). When compared to Sweet Scarlet significant interactions are shown with # *p* < 0.005, * *p* < 0.0005).

**Table 1 molecules-27-00336-t001:** Phenolic compounds content (µg/g dry weight) identified and quantified in the whole freeze-dried grape varieties using HPLC.

Compound	Chemical Group	Flame (Seedless)	Autumn Royal (Seedless)	Sweet Scarlet (Seedless)	Red Globe (Pitted)
Caftaric acid	Cinnamic acid derivatives	48.5 ± 8.30 b	95.8 ± 22.1 a	19.6 ± 1.67 b	54.7 ± 8.80 b
Chlorogenic acid		3.88 ± 0.31 a	3.64 ± 0.53 a	5.06 ± 0.82 a	6.24 ± 1.31 a
Caffeic acid		10.6 ± 3.22 bc	5.49 ± 1.69 c	16.7 ± 1.47 ab	20.2 ± 3.61 a
*p*-Coumaric acid		1.64 ± 0.13 a	1.98 ± 0.11 a	1.74 ± 0.32 a	1.98 ± 0.08 a
Ferulic acid		4.88 ± 0.48 b	4.83 ± 1.09 b	11.1 ± 0.61 a	6.07 ± 0.93 b
Total		69.5	111.7	54.2	89.2
Protocatechuic acid	Hydroxybenzoic acid	18.5 ± 0.43 b	33.9 ± 2.21 a	36.2 ± 5.89 a	31.9 ± 5.71 a
Total		18.5	33.9	36.2	31.9
Catechin	Flavan-3-ols	13.6 ± 0.87 b	25.8 ± 3.22 b	78.9 ± 9.76 a	18.3 ± 2.74 b
Epicatechin		8.94 ± 0.64 b	13.9 ± 1.93 b	41.1 ± 4.34 a	7.40 ± 0.52 b
Epigallocatechin		0.89 ± 0.04 b	2.33 ± 0.66 a	0.73 ± 0.25 b	0.20 ± 0.07 b
Epicatechin gallate		3.29 ± 0.17 bc	12.5 ± 2.02 a	0.74 ± 0.09 c	7.65 ± 2.32 b
Total		26.7	54.5	121.5	33.6
Rutin	Flavonols	ND	ND	ND	ND
Quercetin		2.45 ± 0.05 b	3.05 ± 0.18 b	3.77 ± 0.46 b	24.9 ± 2.55 a
Iso-quercetin		4.02 ± 0.94 b	26.4 ± 3.17 a	26.8 ± 0.80 a	6.49 ± 0.48 b
Myricetin		˂LOD	22.2 ± 2.07 b	4.72 ± 0.59 c	54.2 ± 10.2 a
Kaempferol		ND	ND	ND	ND
Genistein		ND	ND	ND	ND
Total		6.47	51.7	35.3	85.6
Apiginin	Flavones	ND	3.27 ± 0.56 b	2.78 ± 0.22 b	19.7 ± 3.26 a
Total		-	3.27	2.78	19.7
Resveratrol	Stilbenes	3.30 ± 0.81 b	7.23 ± 2.42 b	7.30 ± 2.11 b	13.9 ± 2.87 a
Total		3.30	7.23	7.30	13.9
Delphinidin chloride	Anthocyanins	5.79 ± 0.51 d	53.1 ± 10.9 c	96.4 ± 6.16 b	139.6 ± 7.27 a
Cyanidin glucoside		4.32 ± 1.68 b	˂LOD	16.8 ± 4.09 b	39.3 ± 7.62 a
Pelargonidin chloride		˂LOD	9.11 ± 0.46 a	1.43 ± 0.14 b	1.06 ± 0.22 b
Petunidin 3-*O*-glucoside		0.73 ± 0.08 c	11.1 ± 1.30 a	2.51 ± 0.25 c	5.38 ± 0.68 b
Delphinidin 3-*O*-glucoside		4.50 ± 0.34 c	11.3 ± 0.10 a	4.47 ± 0.20 c	5.58 ± 0.64 b
Total		15.3	84.6	121.6	190.9

Innamic acid derivatives quantified at 325 nm; Hydroxybenzoic acids, Flavan-3-ols, Flavonols, and Flavones at 205 nm; Stilbenes at 306 nm; Anthocyanins at 520 nm; ND: Not detected; LOD: below limit of detection. Values associated with different letters in the same row are significantly different at *p* ≤ 0.05 using Tukey HSD test.

**Table 2 molecules-27-00336-t002:** Fatty acid methyl esters (FAME) content (µg/g fresh weight) in grape varieties (flame, autumn royal, sweet scarlet, red globe pitted, and red globe seeds) using GCMS.

Fatty Acid Methyl Ester/Volatile Compound	Flame (Seedless)	Autumn Royal (Seedless)	Sweet Scarlet (SEEDLESS)	Red globe (Pitted)	Red Globe Seeds
Methyl myristate C14:0	3.09 ± 0.49 d	6.88 ± 1.64 bc	8.43 ± 1.04 ab	4.16 ± 0.08 cd	10.1 ± 0.68 a
Methyl pentadecanoate C15:0	1.07 ± 0.26 b	1.79 ± 0.15 b	1.79 ± 0.20 ab	1.07 ± 0.10 b	2.66 ± 0.53 a
Methyl palmitate C16:0	188 ± 25.0 c	321 ± 28.5 b	337 ± 38.0 b	307 ± 3.12 b	720 ± 41.8 a
Methyl palmitoleate C16:1	3.25 ± 0.43 b	0.84 ± 0.16 c	2.56 ± 0.53 b	3.31 ± 0.08 b	11.1 ± 0.93 a
Methyl stearate C18:0	17.3 ± 1.83 c	46.6 ± 1.75 bc	74.9 ± 9.30 b	39.5 ± 0.94 bc	257 ± 30.0 a
Methyl elaidate C18:1 trans	44.1 ± 1.74 c	47.1 ± 8.89 c	44.8 ± 6.65 c	361 ± 19.8 b	868 ± 114 a
Methyl oleate C18:1 cis	4.17 ± 0.68 c	8.68 ± 1.43 bc	14.2 ± 1.12 b	13.1 ± 0.60 b	28.4 ± 3.9 a
Methyl linoleate C18:2	323 ± 22.0 b	580 ± 53.1 b	859 ± 33.4 b	584 ± 28.3 b	4493 ± 674 a
Methyl arachidate C20:0	11.4 ± 0.54 d	20.4 ± 0.70 b	45.7 ± 0.49 a	20.8 ± 0.52 b	13.5 ± 0.26 c
Methyl α linolenate C18:3	139 ± 24.2 c	305 ± 13.2 a	248 ± 24.8 b	273 ± 19.4 ab	65.3 ± 2.68 d
Methyl behenate C22:0	17.7 ± 0.78 c	28.6 ± 3.04 b	33.3 ± 5.81 b	48.1 ± 3.11 a	14.0 ± 0.41 c
Methyl tricosanoate C23:0	2.43 ± 0.45 c	20.5 ± 2.00 b	3.24 ± 0.85 c	26.2 ± 3.51 a	1.17 ± 0.27 c
Methyl lignocerate C24:0	10.0 ± 1.99 bc	10.5 ± 1.33 bc	17.4 ± 2.37 b	65.4 ± 5.26 a	6.78 ± 0.54 c
Hexacosanoic acid C26:0	26.4 ± 4.72 a	15.4 ± 2.50 b	23.6 ± 3.30 ab	32.6 ± 2.71 a	16.2 ± 2.80 b
Hepta 2,4 dienoic acid	24.3 ± 3.59 bc	12.8 ± 2.17 d	34.1 ± 2.17 a	31.3 ± 0.04 ab	18.4 ± 3.42 cd
Stearaldehyde, dimethyl acetal	22.6 ± 1.46 c	39.8 ± 6.27 b	19.8 ± 3.93 c	70.6 ± 8.08 a	ND
Saturated	277.4	471.7	545.4	544.8	1041.4
Unsaturated	513.5	941.6	1168.6	1234.4	5465.8
Ratio USFA/SFA	1.85	2.00	2.14	2.27	5.25
other	46.9	52.6	53.9	101.9	18.4

Values associated with different letters in the same row are significantly different at *p* ≤ 0.5 using Tukey HSD test.

## Data Availability

Not applicable.

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
