# Peer review of "Potential Associations among Bioactive Molecules, Antioxidant Activity and Resveratrol Production in *Vitis vinifera* Fruits of North America"

_molecules, 2022, doi:10.3390/molecules27020336_

Round 1

Reviewer 1 Report

The paper was well-organized for methodology and analytical approach. Moreover  it appear interesting for the new knowledge concerning the properties of the studied Vitis vinifera varieties. The paper is useful for publication after a minor revision, as follows:

The English form can be checked  and improved.

Lines 31-34. It is advisable to update the production and marketing data to 2020.

Line 82. Delete brackets before and after ‘Red Globe’ and specify that the content is 4 times higher in comparison to Autumn Royal and to other samples.

Line 102. Figures must be progressively numbered following the discussion. So change figure 4  A, B with figure 1 A, B and also check the right sequence of figure and tables with the text.

Lines 104-109. In discussion, make reference to compounds using the same denomination reported in the table.

Author Response

Following include the reply to reviewer 1

The paper was well-organized for methodology and analytical approach. Moreover  it appear interesting for the new knowledge concerning the properties of the studied Vitis vinifera varieties. The paper is useful for publication after a minor revision, as follows:

The English form can be checked  and improved.

Thank you very much for your comments. We double checked the language and the spelling of the contents.

We agree with the reviewers suggestions and addressed them as below. Thank you very much!

Lines 31-34. It is advisable to update the production and marketing data to 2020.

Updated as suggested.

Line 82. Delete brackets before and after ‘Red Globe’ and specify that the content is 4 times higher in comparison to Autumn Royal and to other samples.

Changed as suggested.

Line 102. Figures must be progressively numbered following the discussion. So change figure 4  A, B with figure 1 A, B and also check the right sequence of figure and tables with the text.

Agree with the suggestion. Figure orders were changed  and an updated figure was added to the list.

Lines 104-109. In discussion, make reference to compounds using the same denomination reported in the table.

Changed as suggested in the discussion of results.

Reviewer 2 Report

  • The authors tried to report an interesting, and timely examination of the antioxidant activity of the selected fresh berries of North American with the extended study constitutive stilbene synthase gene expression on resveratrol production.
  • However, manuscript mainly focused to resveratrol for its therapeutic potential. As such, I recommend acceptance after first considering several minor and major suggestions as described below:

Minor suggestions:

  • The work is well planned and should be expected to attract high interest with the MDPI-Molecule’s readership. Unfortunately, the introduction and complete manuscript seems to be vague and quite confusing with use of varieties of grapes. So, I request authors to look carefully and modify it followed by addition of extra paragraph on Trans and Cis Resveratrol also.
  • The Authors are not specific to their figures. The image is not having any Figure Legend or Title. I request authors to have a look for its Figure Legend or Title with proper discussion. This need to be explained briefly.
  • Authors should follow comparison of 4 varieties of grapes but sometimes they compare 5 varieties as shown in figure 2, 3, 4. This should be modified properly and discussed.

Major suggestions:

  • In-vivo /animal studies need to be plan accordingly for to support the potential of work.
  • One Schematic Diagram/ graphical Abstract is required to attract the attention of readers.
  • As such, I recommend acceptance after first considering several minor and major suggestions as described above.

Author Response

Addressing reviewer 2 comments

 The authors tried to report an interesting, and timely examination of the antioxidant activity of the selected fresh berries of North American with the extended study constitutive stilbene synthase gene expression on resveratrol production. However, manuscript mainly focused to resveratrol for its therapeutic potential. As such, I recommend acceptance after first considering several minor and major suggestions as described below:

Thank you very much! We agree with the suggestions

 Minor suggestions:

The work is well planned and should be expected to attract high interest with the MDPI-Molecule’s readership. Unfortunately, the introduction and complete manuscript seems to be vague and quite confusing with use of varieties of grapes. So, I request authors to look carefully and modify it followed by addition of extra paragraph on Trans and Cis Resveratrol also.

Extra paragraph on trans and cis resveratrol was added to the introduction.

The Authors are not specific to their figures. The image is not having any Figure Legend or Title. I request authors to have a look for its Figure Legend or Title with proper discussion. This need to be explained briefly.

Extra information was added to the figures

Authors should follow comparison of 4 varieties of grapes but sometimes they compare 5 varieties as shown in figure 2, 3, 4. This should be modified properly and discussed.

We used only 4 varieties. Three of them are seedless varieties (Flame, Autumn Royal and Sweet Scarlet) and another one include seeds (Red Globe). We analyzsed the variety which include seeds as pitted separating seeds in some instances for comparison. It was revised again and clearly defined in figure legends, results and discussion parts for more clarity.

Major suggestions:

 In-vivo /animal studies need to be plan accordingly for to support the potential of work.

We included information of suggested in vivo and animal studies in the discussion section and the conclusion parts.

One Schematic Diagram/ graphical Abstract is required to attract the attention of readers.

Suggested diagram was added modifying the Figure 1

As such, I recommend acceptance after first considering several minor and major suggestions as described above.

Thank you very much for your review and the suggested changes were addressed accordingly.

Round 2

Reviewer 2 Report

Can be accepted in the present form.